# Pleural Pressure Pulse in Patients with Pleural Effusion: A New Phenomenon Registered during Thoracentesis with Pleural Manometry

**DOI:** 10.3390/jcm9082396

**Published:** 2020-07-27

**Authors:** Elzbieta M. Grabczak, Marcin Michnikowski, Grzegorz Styczynski, Monika Zielinska-Krawczyk, Anna M. Stecka, Piotr Korczynski, Krzysztof Zielinski, Krzysztof J. Palko, Najib M. Rahman, Tomasz Golczewski, Rafal Krenke

**Affiliations:** 1Department of Internal Medicine, Pulmonary Diseases and Allergy, Medical University of Warsaw, Banacha 1A, 02-097 Warsaw, Poland; mgrabczak@vp.pl (E.M.G.); monikazielinska@hotmail.com (M.Z.-K.); drkorczynski@gmail.com (P.K.); 2Nalecz Institute of Biocybernetics and Biomedical Engineering, Ksiecia Trojdena 4, 02-109 Warsaw, Poland; mmichnikowski@ibib.waw.pl (M.M.); astecka@ibib.waw.pl (A.M.S.); kzielinski@ibib.waw.pl (K.Z.); kpalko@ibib.waw.pl (K.J.P.); tomasz.golczewski@ibib.waw.pl (T.G.); 3Department of Internal Medicine, Hypertension and Vascular Diseases, Medical University of Warsaw, Banacha 1A, 02-097 Warsaw, Poland; gstyczynski@wum.edu.pl; 4Oxford Centre for Respiratory Medicine and Oxford Respiratory Trials Unit, Oxford University Hospitals NHS Foundation Trust, Headley Way, Headington, Oxford OX3 9DU, UK; najib.rahman@ndm.ox.ac.uk; 5NIHR Oxford Biomedical Research Centre, University of Oxford, Oxford OX3 9DU, UK

**Keywords:** pleural effusion, pleural manometry, pleural pressure, pleural pressure pulse, thoracentesis

## Abstract

Pleural manometry enables the assessment of physiological abnormalities of lung mechanics associated with pleural effusion. Applying pleural manometry, we found small pleural pressure curve oscillations resembling the pulse tracing line. The aim of our study was to characterize the oscillations of pleural pressure curve (termed here as the pleural pressure pulse, PPP) and to establish their origin and potential significance. This was an observational cross-sectional study in adult patients with pleural effusion who underwent thoracentesis with pleural manometry. The pleural pressure curves recorded prior to and during fluid withdrawal were analyzed. The presence of PPP was assessed in relation to the withdrawn pleural fluid volume, lung expandability, vital and echocardiographic parameters, and pulmonary function testing. A dedicated device was developed to compare the PPP to the pulse rate. Fifty-four patients (32 women) median age 66.5 (IQR 58.5–78.7) years were included. Well visible and poorly visible pressure waves were detected in 48% and 35% of the patients, respectively. The frequency of PPP was fully concordant with the pulse rate and the peaks of the oscillations reflected the period of heart diastole. PPP was more visible in patients with a slower respiratory rate (*p* = 0.008), a larger amount of pleural effusion, and was associated with a better heart systolic function assessed by echocardiography (*p* < 0.05). This study describes a PPP, a new pleural phenomenon related to the cyclic changes in the heart chambers volume. Although the importance of PPP remains largely unknown, we hypothesize that it could be related to lung atelectasis or lower lung and visceral pleura compliance.

## 1. Introduction

The pulse is a fundamental parameter of physical examination. The first device for measuring the pulse rate was developed by a Venetian physician, Santorio Santori, in 1626 [1,2]. Over the centuries, instruments for pulse detection and measurement as well as general knowledge about arterial and venous pulse have markedly improved. It has been demonstrated that heart cyclic contractions generate blood flow in large vessels and can also interact with adjacent organs by passing on pulsations. This is exemplified by the so-called “lung pulse” which was first described as an early ultrasound sign of complete lung atelectasis [3], and later, also reported in patients with pleural effusion [4,5,6,7].

The ability of the lung to re-expand after pleural fluid withdrawal can be reliably assessed by measuring pleural pressure (P_pl_) during thoracentesis. Thus, pleural manometry has been increasingly used to assess local abnormalities of lung mechanics in patients with pleural effusion (PE) [8,9,10]. While performing pleural manometry, we found that the pleural pressure curve showed oscillations resembling the pulse tracing line. Despite an extensive literature search, we found no data on pleural pressure pulsations and their potential significance. We hypothesized that pleural pressure curve pulsations (pleural pressure pulse (PPP)) recorded during pleural manometry were related to the cardiac pulsation transmitted to the atelectatic lung, and then through the PE. The above hypothesis raised a further question, i.e., whether the PPP occurrence is associated with any specific anatomical and pathophysiological conditions (e.g., lung entrapment) or simply reflects heart hemodynamics transmitted to the pleural cavity. Several previous studies have demonstrated that large volume pleural effusion could have an impact on cardiac hemodynamics, resulting in signs and symptoms resembling cardiac tamponade [11,12,13,14,15]. In cases of increased external pressure resulting from PE, the volume of the heart chambers decreases and probably produces lower amplitude pulsations resulting in only faint pulsations transmitted to other organs including the pleural cavity. Considering the above, another hypothesis was formulated assuming that better hemodynamic heart function, moderate amount of pleural effusion, and lower pleural elastance result in more pronounced and better visible PPP.

As these issues have not been evaluated previously, the general objective of the study was to evaluate a new physiological measurement in the pleural space, i.e., the PPP. The specific aims were as follows: (1) To assess how frequently and in which phase of the respiratory cycle the PPP can be observed in patients undergoing therapeutic thoracentesis; (2) to evaluate whether the PPP is strictly related to heart rhythm and changing volumes of heart chambers; and (3) to evaluate conditions associated with the presence of PPP, including cardiac and pulmonary function, pleural fluid volume, and lung expandability.

## 2. Experimental Section

### 2.1. Study Design

This prospective, single center, observational, cross-sectional study was part of a larger research project supported by the National Science Centre, Poland (grant no. 2012/05/B/NZ5/01343). The study protocol was approved by the Institutional Review Board (KB 105/2012) and registered at ClinicalTrials.gov (NCT02192138). Consecutive patients with pleural effusion who had been referred to the Department of Internal Medicine, Pulmonary Diseases and Allergy for therapeutic thoracentesis (TT) were included. The patients underwent pre- and post-thoracentesis pulmonary function testing and echocardiography, as well as P_pl_ measurements during pleural fluid withdrawal. The study conformed to the standards set by the Declaration of Helsinki. All patients signed an informed consent to participate in the study.

### 2.2. Patients

Adult patients with PE occupying at least 1/3 of the ipsilateral hemithorax (in posteroanterior chest radiograph) were enrolled. Other inclusion criteria included no contraindications for TT and general health condition enabling the prolonged procedure of therapeutic thoracentesis. Consecutive patients were included to avoid selection bias.

### 2.3. Echocardiography and Pulmonary Function Tests

Echocardiography was performed by an experienced cardiologist at three time points: one hour prior to, and 3 and 24 h after thoracentesis. Patients were examined in the left lateral and supine position with a Vivid E9 cardiovascular ultrasound system (GE Healthcare, Horten, Norway) equipped with a M5S transducer (1.5–4.6 mHz). Two-dimensional images and Doppler recordings were acquired and stored on a dedicated workstation (EchoPac, GE Healthcare, Horten, Norway) and analyzed offline. Patients with poor image quality or arrhythmia, at the time of the echocardiography, were excluded from echocardiographic evaluation. Cardiac chamber dimensions and selected parameters of systolic and diastolic function of the left and right ventricles were assessed. The left ventricular end diastolic dimension (LVEDD) was measured in the parasternal long-axis view, whereas the right ventricular end-diastolic dimension (RVEDD) was measured in apical four-chamber view. Because of the suboptimal visualization of the endocardium, the left ventricular ejection fraction measured by using biplane Simpson method was not assessed performed. The left ventricular systolic function was assessed using LV fractional shortening (LVFS) (transversal systolic function), mitral annulus plane systolic excursion (MAPSE), and the mean of the lateral and medial mitral annulus systolic velocity (LV TDI S) using the tissue Doppler method (longitudinal systolic function). The left ventricular diastolic function was assessed using the ratio of the early to late pulse wave Doppler velocities of the mitral inflow (E/A) and tissue Doppler early diastolic velocities of the lateral (E’ lat), medial (E’med) part of the mitral annulus, and their mean value (E’ mean). The E/E’ ratio (the ratio of transmitral Doppler early filling velocity to the mean tissue Doppler early diastolic mitral annular velocity) was calculated as a measure of the left ventricular diastolic filling pressure. The right ventricular systolic function was measured using the tricuspid annulus plane systolic excursion (TAPSE). The right ventricular diastolic function was measured using the tricuspid annulus early diastolic velocity (RV E’). The results of Doppler recordings were averaged from 5 consecutive cardiac cycles.

All patients underwent lung function testing including spirometry, body plethysmography, and diffusion capacity for carbon monoxide (DL_CO_) (BodyBox 5500, Medisoft, Dinant, Belgium) one day before, 3, and 24 h after pleural fluid withdrawal. The measurements were performed in accordance with American Thoracic Society and European Respiratory Society recommendations [16,17,18,19].

### 2.4. Thoracentesis and Pleural Manometry

TT was performed in the sitting position under ultrasound guidance. After application of local anaesthesia, a small-bore pleural catheter (Turkel™ Safety System, Covidien, Whiteley Fareham, UK) was inserted into the pleural cavity in the dependent region. The procedure of pleural fluid withdrawal and pleural manometry was performed as described elsewhere [20,21]. Briefly, after careful removal of air with sterile saline, the vertical zero reference point was defined at the level of catheter insertion into the chest. A baseline pleural pressure curve was registered before the beginning of pleural fluid withdrawal. Subsequently, the pleural pressure curve was registered after withdrawal of each 200 mL of pleural fluid up to a total volume of 1000 mL. Pleural pressure curves were recorded during tidal breathing. The duration of pleural pressure registration in each volume point was 60 s. When the volume of the removed fluid exceeded 1000 mL, registrations were performed after removal of each 100 mL. Vital signs and symptoms were registered together with pleural pressure changes. The pleural pressure, as well as other parameters, were continuously processed by data acquisition system and stored in the computer memory for further analysis.

### 2.5. Pleural Pressure Pulse Assessment

A simple photoplethysmographic device (pulse recorder) was built to detect and register the pulse rate during the procedure and to assess whether pleural pressure (P_pl_) curve oscillations were associated with pulse rate. After the placement of the pulsometer on the index finger, the rhythmical changes of blood volume in finger vessels were detected by illuminating the skin with the light from a light-emitting diode (LED) (Figure 1). The amount of transmitted or reflected light was measured by a detector photodiode and the signal was registered parallel to the pleural pressure curves. The signal of pulse rate was compared to waves recorded on the pleural pressure curve to prove their consistency.

A dedicated software was developed to perform a reliable comparative analysis of pulse and pleural pressure oscillations after the completion of therapeutic thoracentesis. Two investigators (EMG and GS) independently evaluated the recorded traces of P_pl_ with regard to the presence of characteristic waves which were classified as visible (waves +), undetectable (waves −), and poorly visible (waves +/−) (Figure 2A–D). Statistical analyses were based on the above three groups of patients but also included subanalyses that compared the following two groups: well visible vs. poorly visible and undetectable waves.

After TT completion, the records of pleural pressure curves were analyzed retrospectively at the following three measurement points that reflected the relative volume of withdrawn pleural fluid: (1) directly after pleural catheter insertion (0% of pleural fluid removed); (2) after removal of 50% of total withdrawn pleural fluid volume; and (3) during the last P_pl_ recording, just before the termination of the procedure (100% of removed pleural fluid volume). The presence or the absence of waves in the first measurement point was assessed in relation to side and volume of PE, ability of lung to re-expand expressed as pleural elastance, vital signs (blood pressure, heart rate, and respiratory rate), hemodynamic parameters measured in echocardiography, and pulmonary function. Variations of waves in subsequent measurement points were also evaluated.

### 2.6. Statistical Analysis

Statistical analysis was performed using Statistica 13.1 (StatSoft Inc., Tulsa, OK, USA) and MedCalc Statistical Software version 13.2.2 (MedCalc Software bvba, Ostend, Belgium). The Kolmogorov–Smirnov test was applied to determine data distribution. Since the majority of data showed non-normal distribution, data were presented as median and interquartile ranges (IQRs, 25th to 75th percentiles). The differences among continuous variables in independent groups were tested using the nonparametric Kruskal–Wallis test and Dunn’s post hoc test. The respective differences in two independent groups were tested using the nonparametric Mann–Whitney U test. The Chi-square test was used to test the differences in terms of categorical variables. Correlations were assessed using Spearman’s rank correlation coefficient. All *p* values were 2-tailed and *p* < 0.05 was considered statistically significant.

## 3. Results

### 3.1. Patient Characteristics

Sixty-two patients treated between September 2015 and November 2018 met the inclusion criteria and underwent TT with pleural manometry. Full records of pleural pressure curve were available for 54 patients. These patients were included in the final analysis. The study group included 32 women and 22 men, median age 66.5 years (IQR 58.5 to 78.7), right and left-sided PE in 28 and 26 patients, respectively. The pre-thoracentesis chest radiograph showed pleural effusion occupying more than one-third, but less than two-thirds, of the entire hemithorax in 19 patients; more than two-thirds, but not the entire hemithorax, in 22 patients; and the entire hemithorax in the remaining 13 subjects. The origins of pleural effusion were as follows: malignant pleural effusion (*n* = 44), nonspecific pleuritis (*n* = 4), tuberculous pleuritis (*n* = 1), heart failure (*n* = 1), and other causes (*n* = 4). The median volume of withdrawn pleural fluid in the entire group was 1800 mL, median baseline P_pl_ and closing P_pl_ were 3.4 (IQR −0.8 to 7.3) and −14.1 (IQR −18.8 to −7.1) cmH_2_O, respectively. Thus, the median pleural elastance calculated for the entire group was 8.5 (IQR 5.6 to 16.7) cmH_2_O/L.

### 3.2. Pleural Pressure Pulse Characteristics and Origin

During the first P_pl_ measurement, well visible and poorly visible waves were detected in 26 and 19 patients, respectively (Figure 2). In nine patients (16.7%) no pulsations on the pleural pressure curve were identified. Waves were more visible in the plateau portion of the pressure curve at functional residual capacity (FRC) (Figure 2A), being present at the end-inspiratory portion of the curve in only a few patients (Figure 2B). The frequency of the waves was concordant with the pulse rate registered by the pulsometer (Figure 3). Peaks on the pulse trace coincided exactly with the nadirs of the low amplitude, high frequency waves on the pleural pressure curve.

Figure 4 presents the comparison of simultaneous recording of the ECG and pulse trace. The peak of pulse waves fell just after the T wave. This suggested that the negative wave of PPP appeared during systole, when the volume of the chambers was the smallest, and that the diastole was responsible for the positive wave of PPP (Figure 5).

### 3.3. Parameters Characterizing Pleural Effusion, Lung Expandability, and Vital Signs in the Studied Subgroups

Basic comparative characteristics of patients with well visible, poorly visible, and undetectable pulsation waves are presented in Table 1.

There were no significant differences in terms of baseline parameters found between the three subgroups. Numerical values of pleural fluid volume, initial pleural pressure, and calculated pleural elastance, in patients with visible PPP, were higher than that in patients with no visible PPP. However, none of the above reached the level of statistical significance (Table 1).

Although both median systolic and mean blood pressure, as well as heart rate (HR), were higher in groups with well visible and poorly visible waves as compared with the subgroup with undetectable PPP, the differences did not reach statistical significance (Table 1). In contrast, the respiratory rate (RR) in patients with well visible waves was lower than in patients with poorly or undetectable pulsations (*p* = 0.008). When the HR/RR ratio (ratio of HR to RR) was calculated and compared between the three subgroups, significant differences were found with the highest HR/RR ratio in the well visible PPP group (3.9, 3.3, vs. 3.1 in the well visible, poorly visible, and undetectable waves group, respectively, *p* = 0.003). It could suggest that the higher the ratio (due to higher HR or lower RR), the PPP could be better visible. Figure 6 shows these relationships for hypothetical patients with the same HR and different RR. On the basis of the above relationships, it might be supposed that in patients breathing with the same frequency, PPP visibility could depend just on HR (Figure 6).

### 3.4. Relation between PPP and Echocardiographic Parameters

Data on differences in echocardiography, spirometry, body plethysmography, and arterial blood gases between patients with well visible, poorly visible, and undetectable pulsation waves are shown in Table 2. Some results were excluded from the analysis because of technical problems, for example, poor image quality.

Nonsignificant differences in RV and LV end diastolic dimension were detected in echocardiography, however, both median LVEDD and RVEDD were smaller in the wave group. The highest values of parameters characterizing right ventricle systolic and diastolic function (TAPSE and RV E’) were found in the subgroup with well visible PPP (*p* = 0.037 and *p* = 0.079, respectively). With the exception of the lateral and medial mitral annulus systolic velocity assessed using tissue Doppler method (LV TDI S lat and LV TDI S med), there were no significant differences in echocardiographic parameters characterizing LV systolic function. Similarly, only one parameter reflecting LV diastolic function (LV TDI E’ med) was significantly higher in patients with visible waves (*p* = 0.008). When echocardiographic parameters were compared in redefined groups (well visible vs. poorly visible and undetectable waves), significant differences in parameters assessing left and RV systolic and LV diastolic function were even more pronounced (Table 3). Although median serum NTproBNP concentration was lower in subgroups with well visible and poorly visible waves, the difference was not statistically significant as compared with patients with undetectable PPP.

### 3.5. Association of PPP with Pulmonary Function and Arterial Blood Gases

There were some differences between the results of pulmonary function tests in patients with well visible, poorly visible, and undetectable PPP (Table 2). The highest FEV_1_% pred. and DL_CO_% pred. values were found in patients with well visible PPP. Even though statistically significant differences in oxygen saturation (SaO_2_) and partial pressure of oxygen (PaO_2_) between patients with well visible, poorly, and invisible oscillations were demonstrated (Table 2), it seems doubtful whether they are clinically relevant. This opinion could be supported by the fact that when the study groups were reclassified, as presented in Table 3, no differences between ABG parameters were found. Nonetheless, in the reclassified groups the patients with visible PPP were still characterized by lower respiratory rate, higher parameters of right and left ventricle systolic function, and LV diastolic function, as well as higher median DL_CO_ (Table 3).

### 3.6. Consistency of PPP during Thoracentesis and Pleural Fluid Withdrawal

The results of consistency analysis of PPP in consecutive measurements are presented in Figure 7. In each subgroup defined by the result of the first measurement, at least half of the patients preserved the same pattern in the following measurement. During the last measurement well visible, poorly visible, and undetectable waves were detected in 17, 22, and 15 patients, respectively (Figure 7).

The presence of well visible waves in the last measurement point was related to significantly lower RR both prior and after procedure termination (*p* = 0.0041 and *p* = 0.0262, respectively), as compared with waves +/− and waves groups. There were no significant relationships between the presence of curve pulsation and pulmonary function and echocardiographic parameters evaluated after the procedure were noted.

## 4. Discussion

Our study shows that pleural manometry performed in patients with PE can reveal low frequency pleural pressure oscillations associated with breathing and also high frequency and low amplitude oscillations which are related to heart hemodynamics. Furthermore, we demonstrated that the nadirs of P_pl_ waves are perfectly matched with the points on the ECG curve (directly after the T wave) which correspond to the smallest ventricular volumes during the cardiac cycle. Therefore, we believe these pleural pressure oscillations are caused by the cyclic changes in the volume of heart chambers during their systolic and diastolic phases. Because the small cyclic changes in P_pl_ are associated with heart hemodynamics, we proposed the term “pleural pressure pulse” to describe this phenomenon. In our opinion, this term is suitable because it suggests not only a causal relationship between the pleural pressure oscillations and the mechanic heart function but also indicates the same frequency of the pleural pressure oscillations and the pulse.

We showed that the presence of the PPP can be registered in more than 80% of patients, including both patients with well visible and poorly visible waves (48.1% and 35.2%, respectively). To the best of our knowledge, this is the first study which has demonstrated the presence of the PPP and analyzed its origin. It must be admitted, however, that PPP could have been seen on the pleural pressure curves presented in two earlier publications. These include a graph presented in the paper by Boshuizen et al. [22], in which PPP is clearly visible, and also a screenshot of pleural manometry included in the review article by Feller-Kopman [23].

We believe that one of the most important findings of our study is the observation on the significant differences in the RR and the heart to respiratory rate ratio between the above groups. On the basis of the significantly lower RR and the significantly higher HR/RR ratio in patients with well visible PPP, it could be hypothesized that the PPP would be visible in all patients with adequately low RR and high HR/RR ratio. Thus, the difference between HR and RR would by a prerequisite for the visibility of PPP. This hypothesis should be verified in future studies which would apply P_pl_ trace registration during slow breathing and breath hold.

The PPP analysis requires a sensitive pleural manometer enabling measurement and registration of instantaneous pleural pressure. Therefore, it could have not been done with a simple water manometer or an overdamped water manometer used in earlier studies [24]. This shows that modern, sensitive, electronic manometers can still provide new data that shed light on pleural pathophysiology and interactions between pleural cavity, as well as lung and heart functions [25]. As an example, the authors of one recent study in which an electronic tracking of pleural pressure was applied suggested that cough during therapeutic thoracentesis could exert a beneficial effect by producing an increase of pleural pressure and preventing an excessive pleural pressure decrease during pleural fluid withdrawal [26]. The advances in technical solutions applied in new pleural manometers have largely contributed to a growing interest in using pleural manometry in patients with pleural effusion [24,27]. Currently, two major directions in the studies on pleural manometry can be distinguished. The first is focused on various pathophysiological phenomena and mechanisms responsible for symptoms in patients with pleural effusion [28]. The second is oriented at the potential clinical applications of pleural manometry, including the prevention of pleural pressure related complications during large volume thoracentesis [29].

Possible clinical applications of the PPP registration and analysis still needs to be established. We cannot exclude that the PPP could prove useful in clinical practice. This supposition is based on somewhat similar observations of the “lung pulse”. This phenomenon was first described by Lichtenstein et al. [3] as pulsations of the pleural line synchronized with the heart cycles (instead of lung sliding) in single lung intubated patients and in healthy volunteers during the breath hold and apnoea. It has been shown that the lung pulse could be a reliable marker of complete lung atelectasis. The phenomenon was explained by a better transmission of heart beats through the atelectatic lung. The lung pulse has also been observed in the M-mode chest ultrasonography in patients with pleural effusion and atelectatic lung [4,5,6,7]. On the basis of our observations, we propose that pleural manometry could be a more sensitive and more accurate measure of pleural pulsations, albeit more invasive, than the M-mode US imaging. As our first study report is based on a retrospective evaluation, further studies involving simultaneous pleural ultrasound imaging and PPP measurements are warranted to test this hypothesis.

Searching for potential clinical applications of the PPP, we performed an extensive analysis of the relationships between the presence or absence of the PPP and numerous clinical parameters, including echocardiographic and pulmonary function data. Having no previous information on the PPP, we classified patients according to the visibility of PPP and analyzed the differences among patients with well visible, poorly visible, and undetectable PPP. Although no significant differences were demonstrated between patients with and without visible PPP in terms of the side of pleural effusion or the amount of pleural fluid, a trend towards a higher volume of withdrawn fluid in patients with well visible and poorly visible waves was observed. This could suggest that a more atelectatic lung is a better conducting medium. Moreover, a trend to a lower median initial P_pl_ and a higher elastance was found in patients with undetectable PPP, suggesting worse lung expandability. Hence, even in the absence of statistical significance, it could be hypothesized that the lack of PPP during breath holding in the baseline P_pl_ measurements suggest significant lung atelectasis and changes in the visceral pleura reflecting its lower compliance and worse expandability. The above hypothesis seems to be concordant with observations of Salamonsen [4] and Leemans [7], who reported a lower amplitude of lung pulsation in M-mode in patients with trapped lung. The lack of statistical significance in our study can be attributed to the fact that this physiological exploratory study was not powered to detect clinical differences such as the presence of trapped lung.

Nevertheless, we found significant differences between the patients with well visible waves and the remaining patients in terms of parameters reflecting LV and RV systolic and LV diastolic function. It can be speculated that higher median LVEDD and RVEDD could be associated with a greater stretch of myocardium and consequently, with an increase in the stroke volume [30]. Better heart contraction, expressed also by TAPSE and LV TDI, could result in more pronounced heart movements, and therefore better visible PPP. Moreover, lower serum concentrations of NTproBNP were observed in well visible and poorly visible waves groups (although the difference was statistically irrelevant). We wondered whether the presence of pleural effusion and trapped lung could have increased RV afterload and consequently, elevated NTproBNP concentration. Although the above was observed in animal studies [15], we have not found any human data supporting this hypothesis.

We are aware of several limitations of this study. First, the total number of patients was relatively small. Due to dyspnea, fatigue, and poor performance status of some patients, high quality echocardiography and pulmonary function data were not available in 20% to almost 40% of patients initially enrolled in the study. Thus, a small number of patients in the studied groups could be responsible for the lack of statistical power in some analyses. Second, the results presented in this paper come from the retrospective analysis of data collected in the frame of a larger project designed to evaluate pathophysiology of large volume thoracentesis. Hence, we have no data on the appearance of the PPP in patients with small volume pleural effusion and only minor lung atelectasis. Third, our classification into the well visible, poorly visible, and undetectable PPP is subjective and observer dependent. However, in this very first study on the PPP, we had no other more explicit tools and methods to classify our findings. To make our classification more reliable, the PPP analysis was performed by two independent observers who further consulted their findings to reach an agreement. Fourth, as already mentioned, we have no data on the PPP visibility during the voluntary breath hold which could probably shed more light on this new phenomenon. This issue is certainly worth considering in the context of the new prospective studies.

## 5. Conclusions

To conclude, our study demonstrates a new phenomenon termed pleural pressure pulse. The PPP represents low amplitude, cyclic pleural pressure alterations related to changes in the volume of the heart chambers. The relationships between the visibility of the PPP and a low respiratory rate, as well as the high heart to respiratory rate ratio, suggest that the detectability of the PPP largely depends on the above variables. Although the importance of the PPP monitoring remains largely unknown, we hypothesize that its appearance in the baseline P_pl_ measurement during the large volume thoracentesis suggests significant lung atelectasis or lower lung/visceral pleura compliance and worse lung expandability.

## Figures and Tables

**Figure 1 jcm-09-02396-f001:**
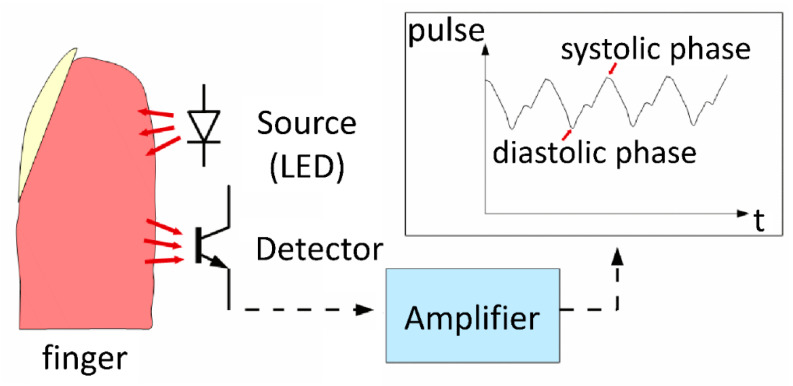
Diagram of engineered finger pulse recorder.

**Figure 2 jcm-09-02396-f002:**
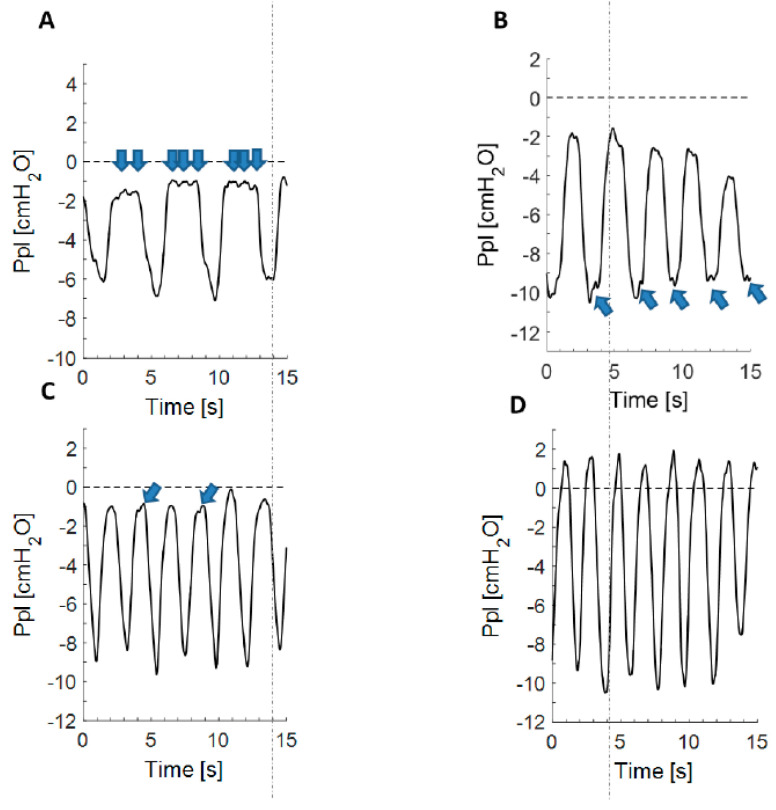
Exemplary pleural pressure curves representing study subgroups. (**A**) Pleural pressure curve with well visible waves (waves +) during end-expiratory phase; (**B**) Pleural pressure curve with well visible waves (waves +) during end-inspiratory phase; (**C**) An extract of pleural pressure curve with poorly visible waves (waves +/−); (**D**) An extract of pleural pressure curve with undetectable waves (waves −). Pleural pressure pulse is marked with arrows.

**Figure 3 jcm-09-02396-f003:**
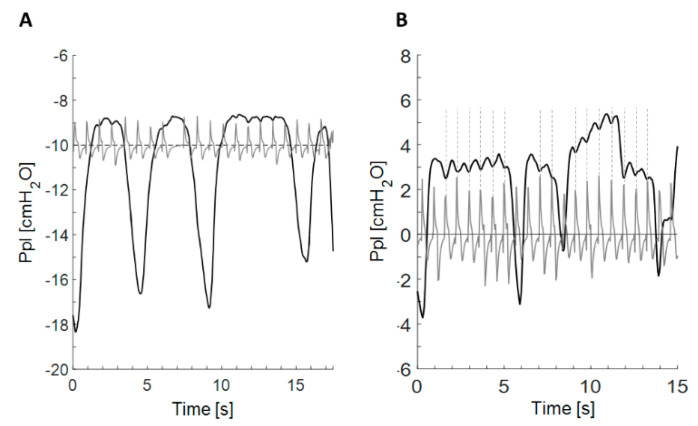
Comparison of pleural pressure curve pulsation with pulse rate recorded by self-developed device. (**A**) Simultaneous record of pleural pressure and pulse curve presenting coincidence of peaks of pulse waves with the most negative point of pleural pressure waves; (**B**) Vertical lines were added to document the coincidence of the peak of pulse trace with the most negative point of pleural pressure oscillations seen on the end-expiratory plateau of the pleural pressure curve.

**Figure 4 jcm-09-02396-f004:**
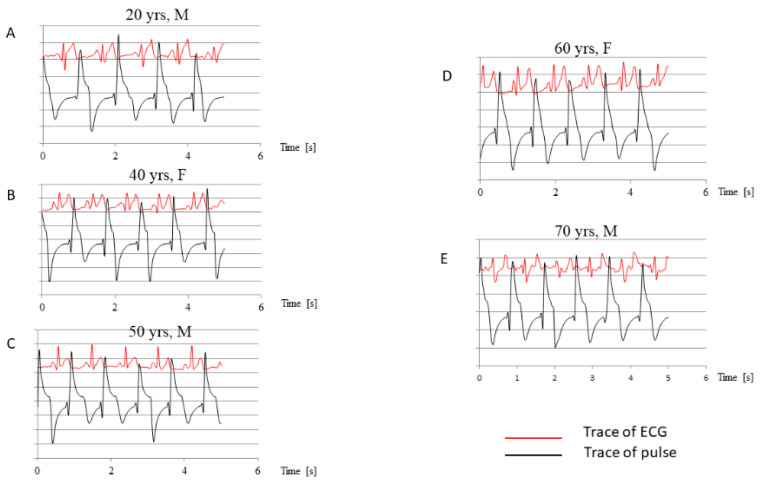
Comparison of simultaneous pulse and ECG trace record. ECG, electrocardiogram; F, female; M, male; yrs, years; (**A**–**E**) panels show the examples of simultaneous traces of ECG and pulse recorded in different patients.

**Figure 5 jcm-09-02396-f005:**
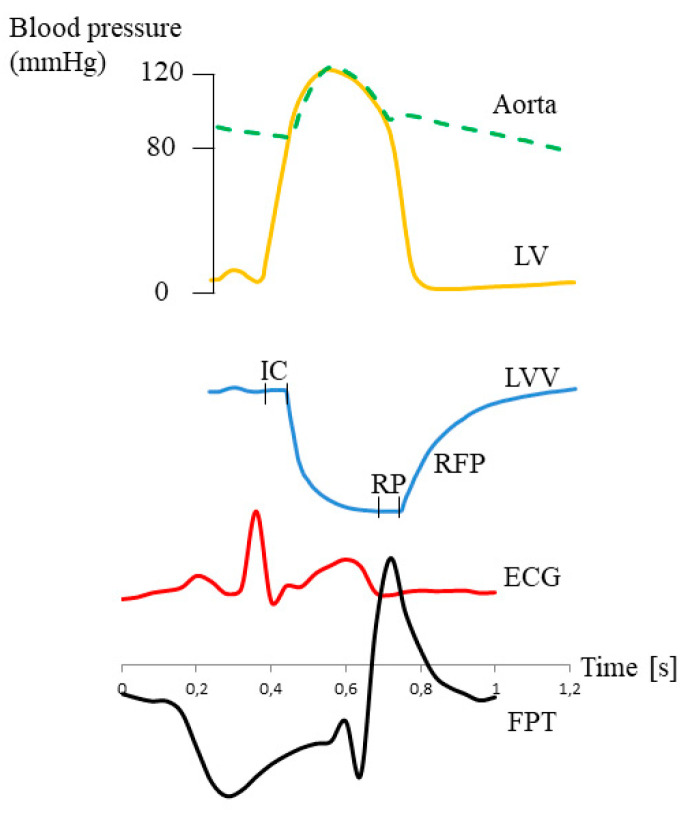
Presumptive relation between pleural pressure pulse (PPP) and the changes in heart ventricle volume. Figure modeled on a figure from G. Ross’s book *Essentials of Human Physiology*, Second Edition 1984, Year Book Medical Publishers Inc., London. Yellow line, pressure in left ventricle (LVP); blue line, left ventricle volume (LVV); IC, isovolumetric contraction; RP, relaxation period; RFP, rapid filling phase; red line, ECG tracing; black line, finger pulse trace (FPT).

**Figure 6 jcm-09-02396-f006:**
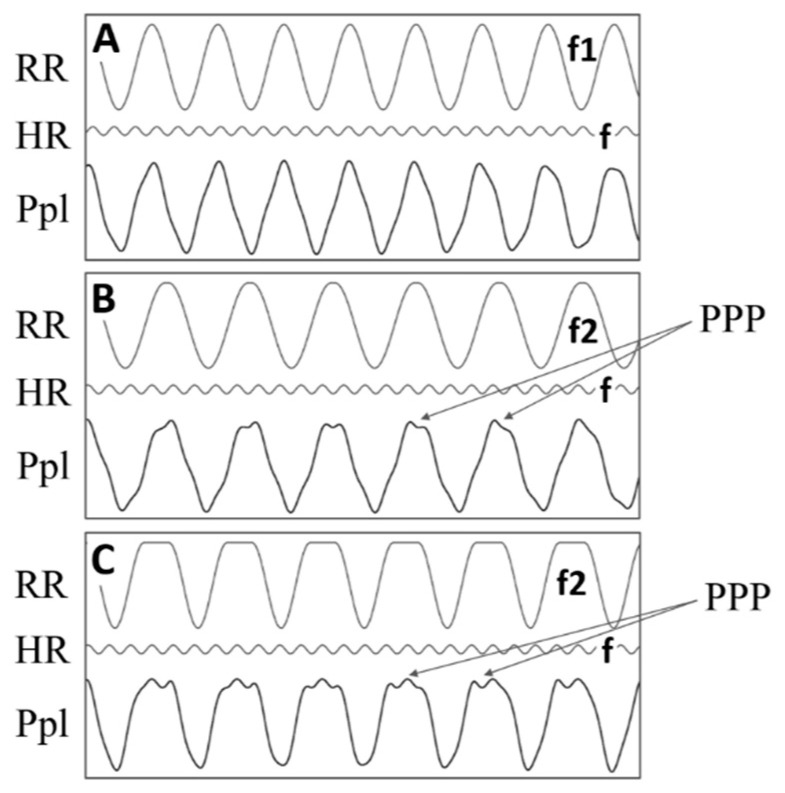
Schematic representation of the effect of superposition of different amplitude and frequency waves on the visibility of the smaller amplitude component. RR, respiratory rate; HR, heart rate; P_pl_, pleural pressure; PPP, pleural pressure pulse. Upper waves in panels (**A**–**C**) represent respiratory derived fluctuations of P_pl_, please note that f2 respiratory frequency (rate) is lower than f1; middle waves in panels (**A**–**C**) correspond to heart derived fluctuations; bottom waves in panels (**A**–**C**) represent superposition of the upper and middle waves. (**Panel A**) Although low amplitude signal (HR) exists, it is invisible on the P_pl_ curve when the ratio of the low and high amplitude signals (reflecting HR/RR) is relatively small (f/f1 = 3.1); (**Panel B**) low amplitude signal becomes visible when the HR/RR ratio becomes higher (f/f2 = 3.9); (**Panel C**) The best visibility of the low amplitude component can be achieved when HR/RR is high and there is a plateau phase on the peaks of the higher amplitude component. Note that although the smaller wave of the higher frequency is the same in all three cases, its influence is not clearly visible when f/f1 = 3.1 and that the visibility of small amplitude signal depends on the frequency ratio of HR and RR only, and it is independent of the specific values of HR and RR considered separately.

**Figure 7 jcm-09-02396-f007:**
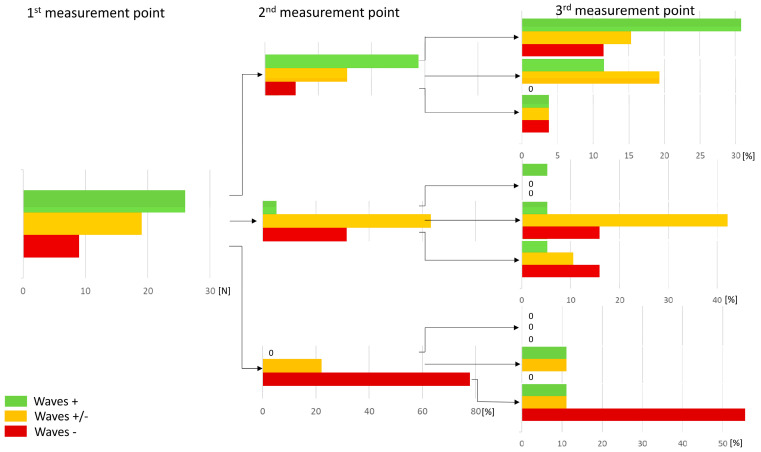
Visualization of pleural pressure curve pulsation during the procedure.

**Table 1 jcm-09-02396-t001:** Comparative characteristics of baseline parameters and data on thoracentesis in patients with well visible, poorly visible, and undetectable pulsations. Data are presented as numbers and percentages or medians and IQRs.

Parameter	Waves +	Waves −	Waves +/−	*p*
Number of Patients (%)	26 (48.1)	9 (16.7)	19 (35.2)	-
Age Years,	63 (57.2–79.7)	72 (58–74)	68 (63.5–77.5)	0.83 *
Gender F/M	16/10	4/5	12/7	0.61 ^#^
BMI kg/m^2^	26 (21.2–28.1)	25.1 (23.1–28.5)	25.7 (22.9–27.3)	0.92 *
SBP mmHg	114 (108–118)	107 (97–114)	114 (100–130)	0.58 *
DBP mmHg	71 (65–75)	66 (58–71)	65 (59.5–72)	0.23 *
MBP mmHg	86 (79.7–92.7)	77 (74.3–85.3)	80 (74.5–90)	0.35 *
HR beat/min	92 (86.5–99.2)	87.6 (81.1–97.7)	80.6 (72.1–94.7)	0.19 *
RR per min	21.7 (18.8–26.9)	30.5 (27.7–34.6)	25.37 (23.5–28.5)	0.008 *
HR/RR	3.9 (3.4–4.9)	3.1 (2.2–3.5)	3.3 (2.9–3.7)	0.003 *
Side of pleural effusion R/L	15/11	3/6	10/9	0.45 ^#^
Distribution of pleural fluid volume assessed in CXR (%) 1/3–2/3 of hemithorax >2/3 of hemithorax The entire hemithorax	9 (34.6) 10 (38.5) 7 (26.9)	4 (44.5) 3 (33.3) 2 (22.2)	6 (31.6) 9 (47.4) 4 (21)	0.94 ^#^
Volume of withdrawn pleural fluid, mL	1910 (1500–2712)	1250 (800–2340)	1700 (1340–2050)	0.25 *
Initial intrapleural pressure, cmH_2_O	2.0 (−0.8–7.7)	0.7 (−1.9–2.8)	4.3 (1.8–6.7)	0.31 *
Pleural Elastance, cmH_2_O/L	8.1 (7.2–13.3)	15.8 (5.5–19.1)	8.4 (4.9–13.3)	0.61 *

BMI, body mass index; CXR, chest X-ray; DBP. diastolic blood pressure; F, female; HR, heart rate; IQR, interquartile range; L, left; M, male; MBP, mean blood pressure calculated as 1/3 SBP + 2/3 DBP; R, right; RR, respiratory rate; SBP, systolic blood pressure; waves +, well visible pleural pressure pulse (PPP); waves +/−, poorly visible PPP; waves −, undetectable PPP; * *p*-value calculated by Kruskal–Wallis test; ^#^
*p*-value calculated by Chi-square test.

**Table 2 jcm-09-02396-t002:** Vital signs, selected echocardiographic, and pulmonary function parameters in 3 subgroups with different degrees of pleural pressure pulse visibility.

Parameter	Waves + (*n* = 26) ^	Waves (*n* = 9) ^	Waves +/− (*n* = 19) ^	*p* *
Blood Gases and Tests				
SaO_2_% ^#^	95.8 (94–96.2) (*n* = 24)	96.1 (94.2–97.2) (*n* = 9)	93.8 (92.9–94.2) (*n* = 16)	0.032
PaO_2_ mmHg ^#^	75.5 (72.3–78.7) (*n* = 24)	82.3 (71.9–83) (*n* = 9)	67.8 (65.8–71.7) (*n* = 16)	0.033
NTproBNP pg/mL	160 (72–338) (*n* = 25)	468 (140–1872) (*n* = 9)	180 (139–575) (*n* = 17)	0.15
Echocardiography				
LVEDD	4.2 (4–4.3) (*n* = 15)	3.1 (3.7–4.5) (*n* = 4)	4.6 (3.9–4.7) (n = 11)	0.67
TAPSE cm	1.9 (1.7–2) (*n* = 16)	1.6 (1.4–1.7) (*n* = 4)	1.6 (1.3–1.9) (n = 11)	0.037
RV E’ cm/s	12 (8.5–15.5) (*n* = 15)	7 (6–9) (*n* = 4)	11 (7–12) (n = 9)	0.079
LV FS	37.8 (26.8–43.7) (*n* = 16)	28.6 (23.6–33.1) (*n* = 4)	41 (33.3–44.1) (*n* = 11)	0.13
LV TDI S lat cm/s	10 (8–12) (*n* = 13)	8.5 (7–9.2) (*n* = 4)	7 (6–8) (*n* = 11)	0.035
LV TDI S med cm/s	8.5 (6.2–9) (*n* = 14)	7 (5.2–8.5) (*n* = 4)	6 (4.5–6.5) (*n* = 11)	0.031
LV TDI S mean cm/s	9 (7.1–10.9) (*n* = 14)	7.7 (6.1–8.9) (*n* = 4)	6 (5.7–7) (*n* = 11)	0.076
LV TDI E’ lat cm/s	9 (8–12) (*n* = 13)	7 (6.5–7.7) (*n* = 4)	8 (6–8.5) (*n* = 11)	0.11
LV TDI E’ med cm/s	8 (6–8.7) (*n* = 14)	6 (5.5–6.2) (*n* = 4)	5 (5–5.5) (*n* = 11)	0.008
LV TDI E’mean cm/s	8.2 (6.6–10.1) (*n* = 14)	6.5 (6–7) (*n* = 4)	6.5 (5.5–7) (*n* = 11)	0.081
E/E’	7.9 (5.9–12.7) (*n* = 13)	10.6 (9.1–11) (*n* = 4)	10.6 (9.1–14.2) (*n* = 11)	0.21
Pulmonary function tests				
TLC L	3.7 (3.5–4.3) (*n* = 20)	4.3 (3.6–5.2) (*n* = 7)	3.1 (2.9–3.8) (*n* = 14)	0.049
TLC% pred	75.6 (65.8–85) (*n* = 20)	79 (66.9–90.3) (*n* = 7)	65 (59.7–72.9) (*n* = 14)	0.12
DL_CO_ ml/min/mmHg	13.4 (11.8–15.2) (*n* = 18)	12.4 (11.6–13.2) (*n* = 6)	10.3 (9.2–11.3) (*n* = 12)	0.022
DL_CO_% pred	58.1 (53.3–63.6) (*n* = 18)	52.1 (44.1–58.5) (*n* = 6)	48.2 (44.2–57.4) (*n* = 12)	0.047
FEV_1_ L	1.3 (1–1.4) (*n* = 22)	1.5 (1.1–1.6) (*n* = 7)	1 (0.8–1.2) (*n* = 14)	0.064
FEV_1_% pred	51.1 (41.7–62.7) (*n* = 22)	53.2 (49.5–62.3) (*n* = 7)	43 (33.9–47.2) (*n* = 14)	0.02
FVC L	1.7 (1.5–2) (*n* = 22)	1.8 (1.5–2.2) (*n* = 8)	1.4 (1.2–1.9) (*n* = 14)	0.24
FVC% pred	55.9 (45.2–70) (*n* = 22)	62.5 (49.7–66.2) (*n* = 8)	51.2 (42.2–54) (*n* = 14)	0.26

All parameters were assessed prior to pleural fluid evacuation. Data are presented as median values and IQRs in parenthesis. Number of patients (n) reported in round brackets. DL_CO_, diffusing capacity of the lung for carbon monoxide; E/E’, the ratio of transmitral Doppler early filling velocity to tissue Doppler early diastolic mitral annular velocity; FEV_1_, forced expiratory volume in 1 s; FVC, forced vital capacity; LVEDD, left ventricular end diastolic dimension; LV FS, left ventricle fractional shortening; LV TDI, tissue Doppler imaging of left ventricle; LV TDI S lat, tissue Doppler systolic velocity of the lateral annulus; LV TDI S med, tissue Doppler systolic velocity of medial annulus; LV TDI S mean, the mean of the lateral and medial mitral annulus systolic velocity using tissue Doppler method; LV TDI E’ lat, tissue Doppler early diastolic velocity of the lateral annulus; LV TDI E’ med, tissue Doppler early diastolic velocity of the medial annulus; LV TDI E’ mean, the mean of the lateral and medial mitral annulus diastolic velocity using tissue Doppler method; RV E’, right ventricle early diastolic velocity; TAPSE, tricuspid annular plane systolic excursion; TLC, total lung capacity; waves +, well visible pleural pressure pulse (PPP); waves +/−, poorly visible PPP; waves −, undetectable PPP. * *p*-value calculated by Kruskal–Wallis test. ^, If number of evaluated patients differed from total number in each subgroup (especially in echocardiographic examination or pulmonary function tests), it was presented in brakes in second line of each row; ^#^, parameters measured in arterial blood sample prior to thoracentesis.

**Table 3 jcm-09-02396-t003:** Comparison of vital signs, pulmonary, and echocardiographic parameters using two different patient classifications.

Parameter	First Alternative Subgroup Division			Second Alternative Subgroup Division		
	Waves + (*n* = 26) ^	Waves − and Waves +/− (*n* = 28) ^	*p* *	Waves + and Waves +/− (*n* = 45) ^	Waves − (*n* = 9) ^	*p* *
Effusion and Pleura						
Volume ml	1910 (1500–2712.5)	1665 (1253.7–2117.5)	0.13	1800 (1350–2300)	1250 (800–2340)	0.21
Initial P_pl_ cmH_2_O	2 (−0.8–7.7)	3.8 (−0.3–6)	0.88	3.6 (−0.1–7.6)	0.7 (−1.9–2.8)	0.19
**Pleural elastance** **cmH_2_O/L**	8.1 (7.2–13.3)	8.7 (5.2–16.9)	0.99	8.2 (6.2–13.4)	15.8 (5.5–19.1)	0.38
Vital Parameters						
SBP mmHg	114 (108–118)	113 (98.5–130)	0.65	114 (105–123)	107 (97–114)	0.31
DBP mmHg	71 (65–75)	65 (58.7–71.7)	0.089	69 (62–74.2)	66 (58–71)	0.48
HR beats/min	92 (86.5–99.2)	83 (74.8–96.9)	0.14	90.3 (77.8–97.5)	87.6 (81.1–97.7)	0.79
RR per min	21.7 (18.8–26.9)	26.7 (24.2–30.7)	**0.010**	24.3 (19.8–28.3)	30.5 (27.7–34.6)	**0.005**
HR/RR	3.9 (3.4–4.9)	3.2 (2.8–3.6)	**0.001**	3.7 (3.2–4.7)	3.1 (2.2–3.5)	**0.030**
Blood Gases and Tests						
SaO_2_%^#^	95.8 (94–96.2) (*n* = 24)	94.1 (92.9–95.7) (*n* = 25)	0.18	94.4 (93.2–95.9) (*n* = 40)	96.1 (94.2–97.2) (*n* = 9)	0.18
PaO_2_ mmHg^#^	75.5 (72.3–78.7) (*n* = 24)	69.8 (66.3–75.7) (*n* = 25)	0.19	72.7 (66.7–76.4) (*n* = 40)	82.3 (71.9–83) (*n* = 9)	0.17
**NTproBNP pg/mL**	160 (72–338) (*n* = 25)	278 (140–606.7) (*n* = 26)	0.081	180(90.2–485.2) (*n* = 42)	468 (140–1872) (*n* = 9)	0.12
Echocardiography						
LVEDD	4.2 (4–4.3) (*n* = 15)	4.3 (3.8–4.7) (*n* = 15)	0.65	4.2 (4–4.6) (*n* = 26)	4.1 (3.7–4.5) (*n* = 4)	0.62
TAPSE cm	1.9 (1.7–2) (*n* = 16)	1.6 (1.3–1.8) (*n* = 15)	**0.019**	1.9 (1.6–2) (*n* = 27)	1.6 (1.4–1.7) (*n* = 4)	0.062
RV E’cm/s	12 (8.5–15.5) (*n* = 15)	9 (7–12) (*n* = 13)	0.052	11 (8–14) (*n* = 24)	8.5 (7–9.2) (*n* = 4)	0.070
LV FS	37.8 (26.8–43.7) (*n* = 16)	35.6 (32.2–43) (*n* = 15)	0.98	38.1 (32.1–44.1) (*n* = 27]	28.6 (23.6–33.1) (*n* = 4)	0.062
LV TDI S lat cm/s	10 (8–12) (*n* = 13)	8 (6–8.5) (*n* = 15)	**0.011**	8 (7–10) (*n* = 24)	8 (8–9) (*n* = 4)	0.73
LV TDI S med cm/s	8.5 (6.2–9) (*n* = 14)	6 (4.5–7) (*n* = 15)	**0.012**	6 (5.5–9) (*n* = 25)	6 (6–8) (*n* = 4)	0.78
LV TDI S mean cm/s	9 (7.1–10.9) (*n* = 14)	7 (5.7–7.5) (*n* = 15)	**0.029**	7 (6–9.5) (*n* = 25)	7.7 (6.1–8.9) (*n* = 4)	0.83
LV TDI E’ lat cm/s	9 (8–12) (*n* = 13)	7 (6–8.5) (*n* = 15)	**0.041**	8 (6.7–10.2) (*n* = 24)	7 (6.5–7.7) (*n* = 4)	0.32
LV TDI E’ med cm/s	8 (6–8.7) (*n* = 14)	5 (5–6) (*n* = 15)	**0.002**	6 (5–8) (*n* = 25)	6 (5.5–6.2) (*n* = 4)	0.65
LV TDI E’ mean cm/s	8.2 (6.6–10.1) (*n* = 14)	6.5 (5.5–7) (*n* = 15)	**0.026**	7 (6–9) (*n* = 25)	6.5 (6–7) (*n* = 4)	0.48
E/E’	7.9 (5.9–12.7) (*n* = 13)	10.6 (9.1–12.6) (*n* = 15)	0.12	9.4 (7.8–13.6) (n = (*n* = 24)	10.6 (9.1–11) (*n* = 4)	0.87
Pulmonary Tests						
TLC L	3.7 (3.5–4.3) (*n* = 20)	3.5 (3–4.1) (*n* = 21)	0.26	3.6 (3–4) (*n* = 34)	4.3 (3.6–5.2) (*n* = 7)	0.15
TLC% pred	75.6 (65.8–85) (*n* = 20)	68.3 (61.1–78.2) (*n* = 21)	0.32	69.8 (61.6–80.5) (*n* = 34)	79 (66.9–90.3) (*n* = 7)	0.27
DL_CO_ ml/min/mmHg	13.4(11.8–15.2) (*n* = 18)	10.9 (9.7–12.8) *(n* = 18)	**0.011**	11.9 (10.6–14.4) (*n* = 30)	12.4 (11.6–13.2) (*n* = 6)	0.92
DL_CO_%pred	58.1 (53.3–63.6) *(n* = 18)	49.7 (42.7–57.5) (*n* = 18)	**0.014**	54.9 (48.6–62.6) (*n* = 30)	52.1 (44.1–58.5) (*n* = 6)	0.56
FEV_1_ L	1.3 (1–1.4) (*n* = 22)	1.1 (0.8–1.4) (*n* = 21)	0.23	1.1 (0.9–1.3) (*n* = 36)	1.5 (1.1–1.6) (*n* = 7)	0.22
FEV_1_%pred	51.1 (41.7–62.7) (*n* = 22)	46.6(38.6–52.3) (*n* = 21)	0.19	46.8 (39–57.9) (*n* = 36)	53.2 (49.5–62.3) (*n* = 7)	0.11
FVC L	1.7 (1.5–2) (*n* = 22)	1.5 (1.2–2) (*n* = 22)	0.37	1.6 (1.3–2) (*n* = 36)	1.8 (1.5–2.2) (*n* = 8)	0.41
FVC%pred	55.9 (45.2–70) (*n* = 22)	52.9 (42.2–62.5) (*n* = 22)	0.25	53.1 (45–66.2) (*n* = 36)	62.5 (49.7–66.2) (*n* = 8)	0.64

All parameters were assessed prior to pleural fluid evacuation. Data are presented as median values and IQRs in parenthesis. Number of patients (n) reported in round brackets. DBP, diastolic blood pressure; DL_CO_, diffusing capacity of the lung for carbon monoxide; E/E’, the ratio of transmitral Doppler early filling velocity to tissue Doppler early diastolic mitral annular velocity; FEV_1_, forced expiratory volume in 1 s; FVC, forced vital capacity; HR, heart rate; LVEDD, left ventricular end diastolic dimension; LV FS, left ventricle fractional shortening; LV TDI, tissue Doppler imaging of left ventricle; LV TDI S lat, tissue Doppler systolic velocity of the lateral annulus; LV TDI S med, tissue Doppler systolic velocity of medial annulus; LV TDI S mean, the mean of the lateral and medial mitral annulus systolic velocity using tissue Doppler method; LV TDI E’ lat, tissue Doppler early diastolic velocity of the lateral annulus; LV TDI E’ med, tissue Doppler early diastolic velocity of the medial annulus; LV TDI E’ mean, the mean of the lateral and medial mitral annulus diastolic velocity using tissue Doppler method; RR, respiratory rate; RV E’, right ventricle early diastolic velocity; SBP, systolic blood pressure; TAPSE, tricuspid annular plane systolic excursion; TLC, total lung capacity; waves +, well visible pleural pressure pulse (PPP); waves +/−, poorly visible PPP; waves −, undetectable PPP. ^#^, parameters measured in arterial blood sample prior to thoracentesis; *, *p* values assed by U Mann–Whitney test. Significant *p* values were bolded. ^, If number of evaluated patients differed from total number in each subgroup (especially in echocardiographic examination or pulmonary function tests), it was presented in brakes in second line of each row.

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
