# Peer review of "Pleural Pressure Pulse in Patients with Pleural Effusion: A New Phenomenon Registered during Thoracentesis with Pleural Manometry"

_jcm, 2020, doi:10.3390/jcm9082396_

Round 1

Reviewer 1 Report

This is a well written report of a newly identified physiologic phenomena, the pleural pressure pulse, which is a cyclical fluctuation in the pleural pressure associated with the cardiac cycle.  The authors identified appropriate patients as a subset of another study and attempted to characterize the etiology along with correlating it with various clinical factors.

Overall this is interesting and well evaluated.  The clinical relevance and importance is questionable, but may prove valuable with further study.  As the authors had significant amounts of data available to evaluate (presumably from the larger study), it sometimes feels like volume of data presented can be overwhelming at times.

Specific comments - 

line 59 - Im not sure the PPP has to reflect cardiac hemodynamics influenced by the PE, it could just reflect unaffected hemodynamics and transmitted changes as you detail later.  It would be interesting to see data from much smaller volume pleural effusions, where atelectasis is less.  I suspect you may still see PPP in many patients.

line 197/198 - the use of 'pulse' (based on the pulsomitor) and pleural pressure pulse becomes confusing in this section.  Maybe just use PPP abbreviation here? 

205 - figure 5, the labels could be cleaned up - RP on diagram, IR in the legend

221-225 - Describing trends that are not statistically significant seems dangerous, though you are careful to be clear

233 - Why the focus on the HR/RR when the difference is driven by the RR?  There is nothing in the presented data to suggest that patients with tachycardia have more visible PPP.  HR was higher in the group that didn't have PPP than in the poorly visible group.  It seems like the focus should be on RR.

I found tables 2-3 to contain way too much information to be easily read.  Why not choose more simple markers of function and hemodynamics?  With so many measurements, one of them is bound to be significant by chance.

306-308 - The numbers in Table 2 make it difficult to break this up into 3 groups, and really hard to make judgements about the data (which is why table 3 is important).  The sat of the non-visible PPP group may be highest - a mean of 96.1 (in 9 patients) is higher 95.8 (in 24 patients), but its really just the same and likely within the error of the measuring instrument.

346 - why focus on HR/RR when everything seems to be driven by RR as above?

369 - albeit more invasive

380-381 -- Ppl and elastance may be higher, but none were close to statistically significant between any of the group lines drawn (at least 3 that I can see), so tough to draw conclusions.

386 - 'likely' seems strong

394-397 -- this is interesting, but doesn't seem within the scope or goals of the paper as it has no relationship to the derivation of the PPP

Author Response

Dear Reviewers,

It was a great pleasure to read your encouraging opinion on our manuscript. We thank you for your insightful review of our manuscript and for your voluntary effort undertaken to improve the manuscript quality. We found your comments and suggestions inspiring and useful. As you suggested, we reduced the number of data presented in the manuscript to make the paper more clear and comprehensible. Below, we enclose our responses to your specific comments and information on the changes made to the original version of the manuscript. These changes are also marked in red in the manuscript file. We hope that these changes are consistent with the intentions of the Reviewers and will increase the chances for publication of the manuscript in Journal of Clinical Medicine.

Reviewer 1

Comments and Suggestions for Authors

This is a well written report of a newly identified physiologic phenomena, the pleural pressure pulse, which is a cyclical fluctuation in the pleural pressure associated with the cardiac cycle.  The authors identified appropriate patients as a subset of another study and attempted to characterize the etiology along with correlating it with various clinical factors.

Overall this is interesting and well evaluated.  The clinical relevance and importance is questionable, but may prove valuable with further study.  As the authors had significant amounts of data available to evaluate (presumably from the larger study), it sometimes feels like volume of data presented can be overwhelming at times.

Specific comments:

line 59 - Im not sure the PPP has to reflect cardiac hemodynamics influenced by the PE, it could just reflect unaffected hemodynamics and transmitted changes as you detail later.  It would be interesting to see data from much smaller volume pleural effusions, where atelectasis is less.  I suspect you may still see PPP in many patients.

We thank the Reviewer for this comment which we fully agree with. Indeed, the expression that “PPP may reflect heart hemodynamics influenced by PE” can be considered awkward and inappropriate. We also feel that PPP may just reflect unaffected hemodynamics and pleural effusion may simply form the window of opportunity to register this phenomenon. The sentence was changed appropriately to the standpoint present above. We share the Reviewer’s opinion that data on PPP in patients with small volume pleural effusion and small lung atelectasis would be interesting. Unfortunately, as our project was focused on patients with large volume pleural effusion undergoing therapeutic thoracentesis, we have no data from diagnostic thoracentesis in patients with small volume of pleural fluid. Based on the Reviewer’s suggestion, we are going to evaluate the presence of PPP in patients with small volume pleural effusion undergoing diagnostic thoracentesis.     

line 197/198 - the use of 'pulse' (based on the pulsomitor) and pleural pressure pulse becomes confusing in this section.  Maybe just use PPP abbreviation here? 

This is a very apt remark. As the suggestion presented by the Reviewer (PPP) also includes the term “pulse”, we decided to change the figure legend to a more descriptive one. Now, it reads: “Vertical lines were added to document the coincidence of the peaks of pulse trace with the most negative points of pleural pressure oscillations seen on the end-expiratory plateau of the pleural pressure curve.” We hope this change is consistent with the Reviewer’s intention.

205 - figure 5, the labels could be cleaned up - RP on diagram, IR in the legend 

We thank the Reviewer for pointing out the error in Figure 5. It has been corrected to make all labels concordant with the abbreviations explained in the legend. To make the figure even more clear, information on the measurements presented with different line colors was added.       

221-225 - Describing trends that are not statistically significant seems dangerous, though you are careful to be clear 

Yes, we absolutely agree with this comment. Therefore, we tried to be cautious, as noted by the Reviewer. As some trends in some parameters seem interesting and may have possible clinical implications, our intention was to attract the Reader’s attention to these parameters and point to the results which may warrant further evaluation. Now, after the Reviewer’s comment, we reduced the text volume which points to trends with no statistical significance to one sentence. This sentence is preceded by a clear statement on the lack of significant difference and followed by a clear information that the presented trends did not reach statistical significance.

233 - Why the focus on the HR/RR when the difference is driven by the RR?  There is nothing in the presented data to suggest that patients with tachycardia have more visible PPP.  HR was higher in the group that didn't have PPP than in the poorly visible group.  It seems like the focus should be on RR.

This is a very pertinent comment we cannot argue with. Indeed, at a glance, RR is the major variable that differs in patients with well poorly and invisible PPP. Based on our observations and calculations it can be concluded that the lower RR, the better visibility of PPP. This can be partly explained by longer end-expiratory plateau that improves PPP visibility. Unfortunately, as the pleural pressure records were analyzed retrospectively, we could not assess whether breath holding improves visibility of PPP in all patients. This is going to be one of the important points in our new study. However, RR seems to be not the only factor affecting PPP. Even though HR was not significantly different in patients with well poorly and invisible waves, our analysis showed that visibility of PPP might be particularly related to HR/RR ratio. This conclusion is derived from our mathematical calculations and definitely requires confirmation in further studies. RR – the denominator in the HR/RR ratio – has to be significantly different entirely for mathematical reasons since HR – the numerator in the HR/RR ratio – is almost the same in all three groups. HR is slightly higher in the wave- group in comparison with the wave+/- group but RR in the wave- group is much higher, and therefore, the HR/RR ratio is smaller. Thus, whether PPP exists or not, it cannot be visible. Figure 6 was designed to present these relationships. This figure shows that visibility of any component with lower amplitude on the resultant wave depends on the frequency ratio of both signals. Hence, even if the small amplitude signal exists, it may not be visible when the ratio of the frequencies is relatively small (f1/f=3.1 in Fig. 6, panel A). In contrast, it becomes better visible when the ratio increases (f2/f=3.9 in Fig. 6, panel B). The best visibility of the smaller component can be achieved when there is a plateau phase on the peak of the higher amplitude component (Fig. 6, panel C). In our patients, this plateau corresponds to the end-expiratory pause, and the greater and smaller components reflect the influence of breathing (higher amplitude signal) and heartbeats (smaller amplitude signal) on pleural pressure, respectively. We would like to stress that the above is a general property of signal superposition, i.e. linear adding of two signals. Note that usually not visual analysis but analysis in the frequency domain [Fourier transform] is used to separate signals of different frequencies. Unfortunately, such an analysis was impossible in our patients because neither breathing nor heartbeat had a constant rate.

To further clarify these relationship we added an additional explanation in the manuscript text and reformulated the title and legend of the Figure 6.

I found tables 2-3 to contain way too much information to be easily read.  Why not choose more simple markers of function and hemodynamics?  With so many measurements, one of them is bound to be significant by chance.

Yes, we realize that both Table 2 and Table 3 include a large number of data. As PPP is a newly discovered phenomenon, our intention was to show this new finding in the context of numerous physiological factors that might be related to its presence. We believe, such an approach might be important to support our considerations on the origin of PPP and potential causes that determine its different visibility in particular patients. However, according to the Reviewer’s suggestion, seven and eight rows presenting less relevant variables were deleted from Table 2 and Table 3, respectively..

306-308 - The numbers in Table 2 make it difficult to break this up into 3 groups, and really hard to make judgements about the data (which is why table 3 is important). The sat of the non-visible PPP group may be highest - a mean of 96.1 (in 9 patients) is higher 95.8 (in 24 patients), but its really just the same and likely within the error of the measuring instrument.

Thank you for the comment. Indeed, albeit the difference was statistically significant, its clinical relevance seems to be highly doubtful. Thus, the statement was reformulated to express doubt in terms of clinical meaning of the statistical differences.

346 - why focus on HR/RR when everything seems to be driven by RR as above?

See our response to the previous comment on RR and HR, please.

369 - albeit more invasive –

Yes, this is certainly true. Unless pleural manometry is a component of therapeutic thoracentesis undertaken due to symptomatic pleural effusion. The comment proposed by the Reviewer was added to the respective sentence.

380-381 -- Ppl and elastance may be higher, but none were close to statistically significant between any of the group lines drawn (at least 3 that I can see), so tough to draw conclusions.

We absolutely agree with the above comment. Nonetheless, our intention was to discuss not only unequivocal results and significant differences, but also to show some vague results and trends which, in our opinion, undoubtedly need further evaluation. We would like to point out that the lack of significance might be, at least partially, associated with the retrospective character of the study, and it might be supposed that a properly designed prospective trial targeted at PPP will shed more light on this phenomenon. We realize that one should be very cautious when presenting and discussing insignificant statistical results and we tried to express this in the discussion. In response to the Reviewer’s remark we further reformulated the text to emphasize the need to be very cautious when interpreting and discussing numerical differences with no statistical significance.   

386 - 'likely' seems strong

Yes, we agree. Only one of the possible explanations of the lack of significant differences was presented. “is likely to be attributed to” has been changed to “might be attributed to…”

394-397 -- this is interesting, but doesn't seem within the scope or goals of the paper as it has no relationship to the derivation of the PPP.

Yes, the Reviewer is right. To explain the presence of the two sentences commented by the Reviewer, we must admit that we were very much intrigued by the origin of PPP. Hence, we performed a comprehensive analysis of the relationship between this new phenomenon and various echocardiographic parameters characterizing heart function. Our intention was to discuss PPP on the broad basis of the left and right ventricle function. However, in response to the Reviewer’s comment, the sentences on the correlations between TAPSE, RV and LV functions were deleted from the revised version of the manuscript.

Reviewer 2 Report

Thank you for submitting this article. I was pleased to receive it as a reviewer.

I have the following questions for you, which I believe, need to be addressed before publication:

First, the statistical analysis should be written according to the recently published guidelines (Hickey GL, Dunning J, Seifert B, Sodeck G, Carr MJ, Beyersdorf F on behalf of the EJCTS and ICVTS Editorial Committees Editor's Choice: Statistical and data reporting guidelines for the European Journal of Cardio-Thoracic Surgery and the Interactive CardioVascular and Thoracic Surgery. Eur J Cardiothorac Surg 2015;48:180-93).

The limitations section should be improved with a better discussion.

Besides, the discussion should be improved with a better search of the literature.

About minor points, there are grammars and typos errors in the text. Please thoroughly check the article.

Good luck with your article, and thanks again for submitting it.

Author Response

Dear Reviewers,

It was a great pleasure to read your encouraging opinion on our manuscript. We thank you for your insightful review of our manuscript and for your voluntary effort undertaken to improve the manuscript quality. We found your comments and suggestions inspiring and useful. As you suggested, we reduced the number of data presented in the manuscript to make the paper more clear and comprehensible. Below, we enclose our responses to your specific comments and information on the changes made to the original version of the manuscript. These changes are also marked in red in the manuscript file. We hope that these changes are consistent with the intentions of the Reviewers and will increase the chances for publication of the manuscript in Journal of Clinical Medicine.

Reviewer 2

Thank you for submitting this article. I was pleased to receive it as a reviewer.

I have the following questions for you, which I believe, need to be addressed before publication:

First, the statistical analysis should be written according to the recently published guidelines (Hickey GL, Dunning J, Seifert B, Sodeck G, Carr MJ, Beyersdorf F on behalf of the EJCTS and ICVTS Editorial Committees Editor's Choice: Statistical and data reporting guidelines for the European Journal of Cardio-Thoracic Surgery and the Interactive CardioVascular and Thoracic Surgery. Eur J Cardiothorac Surg 2015;48:180-93).

We thank the Reviewer for raising the issue of statistical analysis which is a key point of every study. We read with great interest the paper recommended by the Reviewer and we found it easy to understand, convincing and useful for practicing clinicians. We also thoroughly compared the description of statistical analysis included in our manuscript with the EJCTS/ICVTS guidelines. As a result, some information was added to the section Statistical analysis and some changes made in the way of reporting P-value. On the other hand, we think that the major points of the guidelines which refer to reporting observational studies have been addressed in the manuscript. These are as follows:

TITLE AND INTRODUCTION

  • The title of the manuscript is rather informative than “catchy”. We admit that it does not include information on the type of the study, but it is relatively long (18 words) and we doubt whether further increasing of the word count is a reasonable option.
  • We believe that both the working hypotheses and study objectives were clearly presented in the Introduction.

STUDY DESIGN AND OUTCOMES

  • The manuscript has been checked in terms of its consistence with STROBE guidelines and the STROBE checklist can be uploaded on the Editor or Reviewers request.
  • It is probably not possible to clearly predefine outcomes of an observational study which discovered and reported a completely new phenomenon. Nevertheless, the outcomes are consistent with the goals of the study and include the explanation of PPP origin and pointing to the factors related to its appearance.

STATISTICAL METHODS SECTION

  • In the opinion of authors, all statistical methods used in the study were identified. This also refers to the software packages which were used for data analysis.
  • A potential selection bias was minimalized by inclusion of all consecutive patients who met inclusion criteria and agreed to participate in the study. Only 8 (13%) of them had to be excluded from final analysis because the quality and duration of pleural pressure registration was inadequate.
  • The names of the tests used for specific types of analyses were clearly stated.
  • Data distribution was tested using the Kolmogorov–Smirnov test (this information was added to the manuscript) and as the majority of data had non-normal distribution – non-parametric statistical tests were used in data analysis. This applies to both tests used to evaluate the differences between the independent samples and correlation between variables.
  • Missing data were not ignored, but clearly presented, see the numbers of patients in whom a particular variable was measured and subjected for analysis in Table 2 and Table 3. This issue was also discussed in the study limitations section.
  • We believe, the data were presented consistently to the data reporting guidelines

RESULTS SECTION

  • Selective reporting was avoided and this is supported by a large number of variables presented in Table 2 and Table 3. In response to the comment of Reviewer 1, the number of the reported variables was reduced.
  • The way of P-value reporting was changed to present 3 decimal places if P<0.10, and 2 decimal places if ≥0.10.

DISCUSSION SECTION

  • We clearly stated that PPP finding was made by exploratory post hoc data analysis (also see Material and Methods section) and requires further, well-designed studies.
  • Our view on study limitations was presented and this issue was further complemented in the revised version of the manuscript (as suggested by Reviewer 2, see below).
  • The conclusions were based only on findings in the reported study.

The limitations section should be improved with a better discussion.

We thank for this comment, but we have to admit that it is not easy to improve the mentioned section when the comment is very general and does not indicate any specific areas of improvement. We re-thought this section and complemented the discussion of the study limitations. On the other hand, we think that there is no or only very little space for presenting more extensive and detailed discussion of the study limitations because in the revised manuscript this section accounts for 15% of the entire Discussion.   

Besides, the discussion should be improved with a better search of the literature.

            Again, thank you for this critical comment. As we are involved in research on pleural manometry, we make efforts to follow the literature on this topic, and we believe we are aware about the majority of recent publications. However, as the study refers not to a pleural manometry itself but to a new physiological phenomenon (PPP), the number of studies that seem relevant to PPP is rather limited. Five papers published between 2015 and 2020 were added to the reference list and cited in the manuscript. Unfortunately, as we were not provided with any specific advice or suggestion what should be improved, we cannot be sure that the changes made to the text are consistent with the Reviewer’s expectations. We hope that we managed to correctly interpret the Reviewer’s intentions.   

About minor points, there are grammars and typos errors in the text. Please thoroughly check the article.

            The manuscript has been double-checked for typos and grammars. We believe all errors were corrected. 

Good luck with your article, and thanks again for submitting it.

            Thank you for revising our manuscript and for your good luck wish!

Round 2

Reviewer 2 Report

No further comments